# Glyconectin Cell Adhesion Epitope, β-d-Glc*p*NAc3S-(1→3)-α-l-Fuc*p*, Is Involved in Blastulation of *Lytechinus pictus* Sea Urchin Embryos

**DOI:** 10.3390/molecules26134012

**Published:** 2021-06-30

**Authors:** Gradimir Misevic, Iacob Checiu, Octavian Popescu

**Affiliations:** 1Research and Development, Gimmune GmbH, Baarerstrasse 12, 6302 Zug, Switzerland; 2LIBO Medicine Biotechnology Co., Ltd., 78 Dongsheng West Road, Jiangyin 214400, China; 3Gynatal, Assisted Reproduction Center, Str. Protopop George Dragomir 1, 300229 Timisoara, Romania; checiuiacob@gimmune.com; 4Institute for Interdisciplinary Research in Bio-Nano-Sciences, Molecular Biology Center, Babeș-Bolyai University, 400084 Cluj-Napoca, Romania; opopescu.ubbcluj@gmail.com; 5Institute of Biology Bucharest, Romanian Academy, 296 Splaiul Independenței, 060031 Bucharest, Romania

**Keywords:** sea urchin, glyconectins, glycans, cell adhesion, embryogenesis, morphogenesis, cell recognition, glycan–glycan binding, self-assembly

## Abstract

Glycans, as the most peripheral cell surface components, are the primary candidates to mediate the initial steps of cell recognition and adhesion via glycan–glycan binding. This molecular mechanism was quantitatively demonstrated by biochemical and biophysical measurements at the cellular and molecular level for the glyconectin 1 β-d-Glc*p*NAc3S-(1→3)-α-l-Fuc*p* glycan structure (GN1). The use of adhesion blocking monoclonal antibody Block 2 that specifically recognize this epitope showed that, besides Porifera, human colon carcinoma also express this structure in the apical glycocalyx. Here we report that Block 2 selectively immune-precipitate a Mr 580 × 10^3^ (g580) acidic non-glycosaminoglycan glycan from the total protein-free glycans of *Lytechinus pictus* sea urchin hatched blastula embryos. Immuno-fluorescence confocal light microscopy and immunogold electron microscopy localized the GN1 structure in the apical lamina glycocalyx attachments of ectodermal cells microvilli, and in the Golgi complex. Biochemical and immune-chemical analyses showed that the g580 glycan is carrying about 200 copies of the GN1 epitope. This highly polyvalent g580 glycan is one of the major components of the glycocalyx structure, maximally expressed at hatched blastula and gastrula. The involvement of g580 GN1 epitope in hatched blastula cell adhesion was demonstrated by: (1) enhancement of cell aggregation by g580 and sponge g200 glycans, (2) inhibition of cell reaggregation by Block 2, (3) dissociation of microvilli from the apical lamina matrix by the loss of its gel-like structure resulting in a change of the blastula embryonal form and consequent inhibition of gastrulation at saturating concentration of Block 2, and (4) aggregation of beads coated with the immune-purified g580 protein-free glycan. These results, together with the previous atomic force microscopy measurements of GN1 binding strength, indicated that this highly polyvalent and calcium ion dependent glycan–glycan binding can provide the force of 40 nanonewtons per single ectodermal cell association of microvilli with the apical lamina, and conservation of glycocalyx gel-like structure. This force can hold the weight of 160,000 cells in sea water, thus it is sufficient to establish, maintain and preserve blastula form after hatching, and prior to the complete formation of further stabilizing basal lamina.

## 1. Introduction

Morphogenesis during embryonal development can be described as a genetically controlled progression of cell divisions and growth that is accompanied by a multitude of different categories of selective cellular interactions. These processes can be visualized as micro-scale self-assemblies of cells resulting in extremely precise shapes of anatomical structures of adult multicellular organisms. Specific intermolecular binding of the plasma membrane and extracellular matrix glycoproteins and glycolipids are the functional players mediating cell recognition and adhesion. They can be divided into three main classes: protein–protein, protein–glycan, and glycan–glycan [1]. All three are providing the selectivity necessary for cellular recognition and are delivering sufficient binding strength which is sustaining the stable but flexible form of organs and whole multicellular organisms.

Numerous glycoproteins from cadherin, immunoglobulin, integrin, and lectin families, as well as several glycan structural motifs, have been shown to mediate cell recognition and adhesion during embryonal morphogenesis and in adult organisms [1,2,3]. Studies on a variety of species that are possessing different structural complexities lead to the conclusion that cellular interactions are complex and multistep events involving orchestrating action of specific sets of the above-mentioned cell recognition and adhesion molecules [1].

In recent years important research in sea urchin embryogenesis was oriented on molecular networks controlling the expression of certain species of RNAs and/or proteins [4,5,6]. Cell adhesion related to morphogenesis of sea urchin embryos started to be studied many decades ago using in vitro reaggregation of dissociated cells [7,8,9,10]. Subsequently, biochemical aspects of cell adhesion mediated by specific monosaccharides and more complex glycan structures binding to lectins were reported [11,12,13,14,15,16]. For the glycan class of biopolymers, such structure to function related approach is important since it is generally aiming to directly connect developmentally regulated expression of specific glycoproteins and glycolipids to their putative role in cell adhesion and recognition.

Majority of reports on cell adhesion in sea urchin embryos were focused on gastrulation and less on blastulation. Several types of aggregation factors, some of which have been isolated and partially characterized [17,18,19,20,21,22,23,24,25,26,27], as well as more defined proteins such as desmosome cadherins [28,29,30], fibronectin [23,31,32], laminin [32,33], collagen type I, III and IV [34,35,36,37], integrin [38], lectins [39] including Pi-nectin [11,38] and echinonectin [12,40,41,42,43], hyalin [44,45,46,47,48,49,50,51], proteoglycans [40,52,53,54,55,56], and glyconectin type of glycans [15,16], were shown to promote cell–cell adhesion and cell–extracellular matrix adhesion that were implicated in the maintenance of sea urchin embryonal shape and morphogenesis.

During sea urchin blastula formation and proceeding gastrulation three cellular locations of polarized epithelia are involved: (a) lateral cell–cell adhesion via belt desmosome, or zonula adherens, mainly via cadherins, (b) cell–apical lamina adhesion of microvilli plasma membrane and glycocalyx via hyaline and echinonectin glycoproteins, and (c) cell–basal lamina adhesion involving plasma membrane integrin and lectins receptors with diverse classes of extracellular matrix glycoconjugates such as fibronectin, laminin, and proteoglycans. Ectodermal cell attachment to microvilli and apical lamina matrix was previously indicated as adhesion sites that are stabilizing the blastula form prior to the complete formation of the basal lamina [28,57,58,59].

The exact structures of glycans implicated in cell adhesion of sea urchin embryos via glycan–lectin interactions were not determined in most of the in vitro studies [13,14,18,20,39,41,42,43]. Similarly, the exact structures of proteoglycan glycosaminoglycans, that were demonstrated to play a role during embryonal sea urchin cell adhesion, were also not finalized [40,53,54,60]. In addition to these in vitro studies, three types of indirect in vivo experiments have suggested a role for glycans as adhesion molecules during morphogenesis of sea urchins: (a) sea urchin embryos growing in sulfate-deprived sea water were not able to gastrulate and had reduced synthesis of sulfated glycans and proteoglycans [53,54,55,56], (b) intra-blastocoelic microinjected concanavalin-A lectin bound specifically to ectodermal cells and inhibited gastrulation by blocking their attachment with secondary mesenchymal cells [61], and (c) ß-d-xyloside inhibition of glycosaminoglycan attachment to the protein core resulted in abrogation of gastrulation [52,53,56].

Electron microscopical, biophysical, and biochemical studies showed that glycans from proteoglycans, mucins, and glyconectin glycoconjugates represent the most peripheral and highly abundant cell surface glycocalyx and extracellular matrix components in embryos as well as in adult organisms [1,62,63,64,65,66]. It should be therefore expected that the initial adhesive or repulsive cell–cell and cell–extracellular matrix interactions shall be generally guided via glycan–glycan interactions. These shall be followed by less peripheral glycoprotein and glycolipid molecules of lectin, immunoglobulin, cadherin, integrin, and various types of extracellular matrix families of molecules. These classes of cell adhesion molecules were shown to operate via protein–glycan and protein–protein binding in various types of interactions of cells present in adult, embryonal, and tumor tissues [1,5,67,68].

The first direct evidence that Ca^2+^ dependent glycan–glycan interactions of glyconectin 1 mediate cell adhesion in Porifera was demonstrated by atomic force microscopy measurements of binding strength under physiological conditions between individual glyconectin 1 molecules [69]. Intermolecular binding measurements together with structural analyses [70,71] demonstrated that a single pair of sponge glyconectins can hold the weight of 1600 cells in physiological solution through Ca^2+^ dependent intermolecular associations of a highly repetitive and β-d-Glc*p*NAc3S-(1→3)-α-l-Fuc*p* glyconectin 1 epitope structure (GN1) [69,70,71,72,73,74,75,76,77,78,79,80,81,82]. These findings represent the basis for the construction of the presented model of apical lamina glycocalyx cell adhesion in sea urchin blastula.

The monoclonal antibody Block 2 developed against the sponge glyconectin 1 glycan was shown to specifically recognize GN1, β-d-Glc*p*NAc3S-(1→3)-α-l-Fuc*p* adhesion epitope, and to blocks cell adhesion [70,71,72,73,74,75,76,77,78,79,80,81,82]. Immunofluorescence histological and biochemical experiments with the anti-GN1 Block 2 monoclonal antibodies identified similar structures in human colon carcinoma apical glycocalyx [74] and in sea urchin embryos [15,16]. The fact that the novel glycan unit, functioning via a conceptually new adhesive mechanism of polyvalent glycan–glycan interactions, is expressed in the simplest Porifera phylum, and was preserved during evolution in mammalian cancer tissue, led us to test whether this structure is a carcino-embryonal antigen also involved in morphogenesis of sea urchin embryos. If so, during the ontogeny of the multicellular organisms, and in pathological tumor growth, the repetition of phylogenetical processes associated with cell adhesion could be verified on the molecular level for these cohesive glycan structures. Since glycan structures are not directly genetically controlled as proteins are, but rather indirectly via expression and localization of glycosyltransferase generating specific glycan sequences [1,76], structure to function related research on this molecules is not easily connected to genetic and molecular biology research.

Here presented research on cell adhesion function of glyconectin glycans during the first major morphogenetic process of blastulation of sea urchin *L. pictus* embryos is the continuation of the previous studies [15,16,72,73,74]. Transparent *L. pictus* embryos were well suited for in vivo optical microscopy, immune-chemical, and biochemical examination using the anti-GN1 Block 2 monoclonal antibody that specifically recognizes this 1 β-d-Glc*p*NAc3S-(1→3)-α-l-Fuc*p* adhesive disaccharide epitope. Time of expression and localization of this structure in hatched blastula stage, as well as in vivo and in vitro effects of the anti-GN1 monoclonal antibodies in cell adhesion during morphogenesis were performed. The anti-GN1 Block 2 monoclonal antibodies were also used for immuno-purification of a target putative glyconectin glycan carrying the GN1 epitope structure. The cell adhesion function of this immuno-purified glycan was also tested in vitro and in vivo by cell and bead adhesion studies in order to provide experimental evidence about the GN1 glycan epitope function in cell adhesion implicated in blastulation.

## 2. Results

### 2.1. Expression of the GN1 Glycan Cell Adhesion Epitope β-d-GlcpNAc3S-(1→3)-α-l-Fucp during Embryonal Development of Sea Urchin L. pictus

In order to examine, in detail, the expression of the previously reported presence of the GN1 cell adhesion epitope β-d-Glc*p*NAc3S-(1→3)-α-l-Fuc*p* [72,73] during morphogenesis of sea urchin *L. pictus* embryos, glycans were isolated from eggs, morula, blastula, hatched blastula, end gastrula, and prism. Chloroform/methanol precipitation followed by pronase and DNAase digestion, and column chromatography was used as the most appropriate approach to obtain all glycans free of protein (see Material and Methods). Amino acid and monosaccharide composition of total glycans revealed the presence of both *N-* and *O-linked* structures containing fucose, N-acetylglucosamine, N-acetylgalactosamine, glucuronic acid, galactose, and sulfate, and the absence of proteins and peptides.

Immunodot assay of two-fold serially diluted total protein-free glycan fractions obtained from the six developmental stages starting with 0.5 μg glycans per dot was performed with the Block 2 monoclonal antibody previously shown to specifically recognize β-d-Glc*p*NAc3S-(1→3)-α-l-Fuc*p* GN1 epitope (Figure 1 and Figure 2A–C). The semiquantitative immunodot analysis showed that the GN1 epitope structure is expressed in embryos and is strictly regulated during development in the following fashion: absence in eggs, the sharp increase in biosynthesis during blastula formation, reaching maximal expression at hatched blastula and end gastrula, followed by a decrease in later prism stage (Figure 1A). Control experiments confirmed that binding of the Block 2 monoclonal antibodies to doted sea urchin glycans was inhibited at 100 μg/mL concentration of the sponge glyconectin 1 glycan, to which antibody was raised, whereas chondroitin sulfate A, B, and C, keratan sulfate, heparin, heparan sulfate, or hyaluronic acid did not show inhibition at 700 μg/mL concentration (not shown). Besides these control experiments presented in part here, also previous reports showed that the Block 2 monoclonal antibodies did not bind to any of the following classical glycosaminoglycans mentioned above, as well as that binding of this anti-GN1 monoclonal antibodies before pronase digestion and after pronase digestion, where all proteins are removed, is identical [15,16,72,73]. Furthermore, the results presented here together with previously published data showed that purified natural de-sulfated disaccharides, synthetically obtained non-sulfated disaccharides, L—fucose, D—N-acetylglucosamine 3 sulfate, *N-* and *O-linked* glycans from mammalian, and sponge and sea urchin glycoproteins, did not bind to the Block 2 monoclonal antibody and were not capable of inhibiting binding of this antibody to the GN1 epitope of g200 glycan. Contrary only purified and/or synthetically obtained GN1 sulfated disaccharide epitope structure bound to the Block 2 monoclonal antibody and was capable of inhibiting binding of this antibody to g200 [69,70,71,72,73,74,75,76,77,78,79,80,81,82]. These findings revealed strict specificity of the Block 2 monoclonal antibody to the GN1 epitope sequence D-β-d-Glc*p*NAc3S-(1→3)-α-l-Fuc*p* schematically presented in Figure 2A–C [72,73,77,78,79]. Using the unrelated mouse IgG as control additionally confirmed the specificity of Block 2 monoclonal antibodies.

In order to examine the molecular nature of glycans carrying β-d-Glc*p*NAc3S-(1→3)-α-l-Fuc*p* GN1 cell adhesion epitope, protein-free glycans obtained from each developmental stage were separated by gel electrophoresis. One part of the gel was stained with Alcian blue for detection of acidic glycans (Figure 1B), and the other part was electroblotted onto DEAE cellulose and probed with the anti-GN1 Block 2 monoclonal antibodies (Figure 1C). Alcian blue staining revealed the presence of several acidic glycans ranging from the extremely large molecular mass not entering the gel, and then from 580 × 10^3^ D to 2 × 10^3^ D, all of which had developmentally regulated expression.

Based on the monosaccharide analyses of a recovered fraction of the g580 glycans after extraction from gels, as well as by semiquantitative Alcian blue staining of gels, it was estimated that more than 20% of total acidic glycans were represented by the acidic glycan with a molecular mass of 580 × 10^3^ D (g580) as the major component during hatched blastula until end gastrula.

Immunoblotting showed that the only one band, indirectly stained by binding of the anti-GN1 Block 2 monoclonal antibody, was g580 glycan that was previously discovered and partially analyzed [15,16]. The g580 glycan was neither detected by Alcian blue nor by Block 2 staining in eggs but was starting to be expressed at morula and blastula with a maximal presence at hatched blastula and end gastrula stages (Figure 1B,C). Furthermore, a 10-fold molar excess of classical glycosaminoglycan structures were unable to inhibit binding of the antibodies to the electrophoretically separated and electroblotted g580 sea urchin glycans, whereas a two-fold excess of sponge glycans or four-fold excess of total *L. pictus* glycans versus Block 2 completely inhibited its binding to the g580 band (not shown).

Using a semiquantitative approach based on the number of the anti-GN1 Block 2 monoclonal antibodies bound per molecule of g580, it was estimated that one g580 carries about 200 copies of β-d-Glc*p*NAc3S-(1→3)-α-l-Fuc*p* GN1 glycan epitope (Figure 2A–C). Statistical evaluation from 3 different preparation of sea urchin glyconectin glycans revealed that maximal variation error was below 15%. Therefore, the sea urchin g580 glycan contains the same highly polyvalent GN1 adhesion epitope sequence as the sponge g200 glycan [72,73]. Since g580 and g200 have different size and monosaccharide composition (see, next, the immuno-purification section), and since estimated 200 repeats of the GN1 epitope in g580 represent about 20% of the complete sequence, whereas previously determined 100 repeats of the GN1 in g200 is representing about 40% of its complete sequence, the overall structure of these two glycans must be also different and reflected in non GN1 motifs. Although these non-GN1 structures are still remaining to be revealed their imaginary sequences, which are in part based on monosaccharide composition, are also presented as repetitive motifs illustrated with non-colored symbols of monosaccharides in Figure 2B,C.

From the presented biochemical and immunological analyses, as well from the previous partial sequencing, it can be concluded that the GN1 epitope is: (a) different to so far known simpler disaccharide repetitions found in classical glycosaminoglycans [72,73,74], (b) expressed in *L. pictus* embryos in a developmentally regulated fashion, (c) a highly repetitive structure of the g580 acidic glycan, and (d) representing one part of g580 complex and unique structure (Figure 1 and Figure 2).

### 2.2. Immuno-Purification of the Sea Urchin Embryonal g580 Glycan Bearing the GN1 Cell Adhesion Epitope with the Anti-GN1 Block 2 Monoclonal Antibodies

In order to further immuno-purify and examine the possible heterogeneity of g580 glycans in terms of the presence of the GN1 epitope within the same molecular weight range, the protein-free fraction of hatched blastula glycans were immunoprecipitated with the four-fold molar excess of the anti-GN1 Block 2 monoclonal antibodies coated to carboxylated beads. After washing of non-bound glycans, beads with bound glycans were treated with pronase in order to digest the antibody. Unrelated mouse IgG-beads were used as a control. Bound glycans were then separated by gel filtration from amino acids and analyzed by polyacrylamide gel electrophoresis and immunoblotting after gel electrophoresis. Staining of gels with Alcian blue revealed that a single glycan of apparent M_r_ = 580 × 10^3^ D was specifically immunoprecipitated with the Block 2 monoclonal antibody (Figure 1B). After separation of immune-purified glycans by electrophoresis and electroblotting to DEAE cellulose, immuno-decoration with the Block 2 monoclonal antibodies marked the same g580 band (Figure 1C). Immunoadsorption of blotted gels of not bound glycans to the anti-GN1 Block 2 monoclonal antibody did not contain any g580 (not shown). Moreover, control beads did not show detectible adsorption of any glycans. Thus, g580 glycan carries the GN1 epitope structure identical to the adhesion glycan epitope originally found in the g200 glycan of the sponge adhesion glyconectin 1.

The monosaccharides and amino acids composition of g580 hatched blastula glycans were determined by colorimetric reactions, high pressure liquid chromatography, and gas chromatography as described in Material and Methods. Results showed 10% of fucose, 25% of N-acetylglucosamine, 25% of glucuronic acid, and 3% galactose, and 900 moles of sulfate per g580. Less than one mole of asparagine and serine were present per mole of g580. The content of other detectable amino acids was five times lower than that of asparagine. Considering the apparent molecular mass of g580 and the molecular mass of monosaccharides, the compositional analyses of g580 indicated that the majority of N-acetylglucosamine is sulfated and that about 1/3 of them are engaged in linkage with 2/3 of fucose in the 200 repeats of β-d-Glc*p*NAc3S-(1→3)-α-l-Fuc*p* GN1 epitope. Thus, the g580 glycans from hatched blastula are essentially free of proteins with not yet determined linkage to the protein. These results also confirmed the initial estimate reported above that there are about 200 copies of the GN1 epitope within the g580 molecule and that g580, additionally, has the most likely repetitive sequences containing galactose and glucuronic acid that are different from the non GN1 parts of g200 sequences found in sponge glyconectin 1.

In accordance with previously published results for not immunopurified g580 isolated from non-hatched blastula and gastrula [15,16], here reported immunopurified g580 from previously not examined stage of hatched blastula revealed that also this molecule cannot be degraded by glycosidases specific for any known glycosaminoglycan structures. Therefore, g580 has a complex glyconectin type acidic glycan structure that contains the highly repetitive GN1 adhesion epitope and can be generally characterized by a large size, high content of fucose, sulfated N-acetylglucosamine, and glucuronic acid.

### 2.3. Immuno-Histological Localization of the GN1 Glycan Epitope in the Apical Lamina and Golgi of Sea Urchin Hatched Blastula Embryos with the Anti-GN1 Block 2 Monoclonal Antibodies

#### 2.3.1. Confocal Microscopy

In order to associate the obtained biochemical data of developmentally regulated expression of the GN1 glycan epitope with its possible cell adhesion function, the anatomical site of GN1 in *L. pictus* sea urchin hatched blastula was examined by confocal immunofluorescence microscopy with the anti-GN1 Block 2 monoclonal antibody. Embryos were fixed and permeabilized using drop-wise addition of cold methanol followed by rehydration and incubation with 100 μg/mL of the anti-GN1 Block 2 monoclonal antibody as described in Material and Methods. Whole mounts of hatched blastula embryos were indirectly stained with fluorescently tagged secondary antibodies and examined by laser scanning confocal microscopy using epiluminescence and transmitted illumination (Figure 3A–E). A sequence of 30 composite images derived from fluorescence and transmitted light was taken in steps of 0.6 μm of *z*-axis with 63 × objective. Three images that are including *z*-axis steps covering apical lamina and interior of the cell are presented in Figure 3A–C, together with the complete three-dimensional reconstruction shown in Figure 3D. Labeling of two structures was observed. The first intensively stained one had about 0.5 µm diameter and was located within the fuzzy appearing layer of the apical lamina as visualized by transmitted light. In the *z*-axis this structure appeared in about 1 µm thick single layer outside of the main cell body and separated from one another by about 0.5 µm. The second structure type was inside of cells, it had a diameter of about 3–5 µm, and only one of them was present per cell (Figure 3A–D).

The specificity of staining with 100 μg/mL Block 2 was demonstrated by the complete inhibition of the antibody binding with 70 μg/mL of purified g580, or the original antigen, sponge glyconectin 1 β-d-Glc*p*NAc3S-(1→3)-α-l-Fuc*p* epitope structure. A mixture of commercially available glycosaminoglycans (chondroitin sulfate A, B, and C, keratan sulfate, heparin, heparan sulfate, and hyaluronic acid) at concentrations of 700 μg/mL did not inhibit binding of antibodies to the embryo sections (not shown). Since, also, the control nonimmune mouse IgG did not show any labeling and that hatched embryos were examined without fertilization membrane, it could be concluded that microvilli with associated apical glycocalyx matrix (the first labeled layer) and Golgi structures (the second labeled layer) display the GN1 epitope determinants. These results indicated that *L. pictus* g580 carrying the GN1 epitopes is synthesized during blastulation and secreted into the apical lamina.

#### 2.3.2. Immunogold Electron Microscopy

The ultrastructural localization of the glycan epitope recognized by the anti-GN1 Block 2 monoclonal antibody was also performed using indirect immunogold labeling technique of ultra-thin sections (20–50 nm) of hatched blastula embryos (see Materials and Methods). The Block 2 monoclonal antibody bound exclusively to the microvilli and glycocalyx associated layer of the apical lamina that was free after hatching of fertilization membrane (Figure 3F,G), and to the Golgi complex where it is synthesized (Figure 3H). The size of microvilli and apical lamina glycocalyx labeled regions obtained by electron and confocal microscopies were as expected nearly identical. The same was the case with the labeling of Golgi complex.

The control experiments showed that immunogold labeling with Block 2 could be completely inhibited with the sea urchin g580 and sponge glyconectin 1, as was also demonstrated in light microscopy (not shown). In addition, as in biochemical analyses and confocal microscopy control experiments, the mixture of glycosaminoglycans did not constrain antibody binding to embryo sections (not shown).

Although differential interference optics in confocal microscopy analyses did not reveal clearly visible microvilli and a flagellum, just a fuzzy apical lamina layer, the size, abundance, and localization of the labeled structures in this layer correlated well with immunogold electron microscopy labeling of glycocalyx and microvilli in the apical lamina (Figure 3), and with the expected sizes of self-associated g580 structures (Figure 2). These data collectively provided evidence that the GN1 epitope of g580 is present in microvilli associated glycocalyx of the apical lamina.

Based on here reported ultrastructural and biochemical analyses, schematic drawings of the GN1 epitope localization at the molecular and cellular level of hatched blastula were constructed. This model summarizes obtained results that the g580 glycan carrying repetitive GN1 adhesion epitope is highly abundant in glycocalyx and serves as the anchoring points to microvilli and flagella (Figure 2D,E, Figure 3 and Figure 4).

### 2.4. Hatched Blastula Cell Adhesion

Biochemical data provided evidence that the GN1 epitope is present in a highly polyvalent form in g580. Therefore, it should be expected that it can mediate cell adhesion in sea urchin embryos. In order to directly measure such adhesive function, g580 promoted reaggregation of hatched blastula ectodermal cell, g580 coated bead aggregation, and inhibition of hatched blastula cell adhesion by the anti-GN1 Block 2 monoclonal antibodies was performed.

#### 2.4.1. Enhancement of Blastula Cells Reaggregation by the Glyconectin g580 Acidic Glycan

The involvement of the GN1 glycan epitope in cell adhesion of *L. pictus* sea urchin ectodermal cells from hatched blastula was directly tested by measuring the enhancement of cell aggregation by g580 immunopurified from the same stage and by the sponge glyconectin 1, both carrying over 100 repeats of β-d-Glc*p*NAc3S-(1→3)-α-l-Fuc*p* GN1 epitope structure. Dissociated hatched blastula cells were prepared by treating embryos with Ca^2+^ and Mg^2+^ free artificial sea water (CMF-ASW) as described under Material and Methods. After washing by CMF-ASW, 10^5^ dissociated cells, were incubated in 100 μL filtered sea water (FSW), containing calcium and magnesium ions, with either 150 μg/mL g580, or 150 μg/mL of sponge glyconectin 1. The basal level of reaggregation was enhanced after 10 min using the slow rotation of the cultures at 16 °C (Figure 5A,B) and, also, at +4 °C (not shown). In the absence of highly polyvalent GN1 epitopes, cell adhesion was gradually achieved after 1 h at 16 °C due to synthesis and secretion of endogenously produced g580 (Figure 5C). In the absence of Ca^2+^ and Mg^2+^ cell did not aggregate after 1 h (absence of adhesion was as in Figure 5D). The observed level of cell aggregation promoted by the GN1 bearing glyconectins was, as expected, increasing with the increasing concentration of these molecules. The reaggregation experiments demonstrated that the GN1 glycan epitope in his highly polyvalent form in both sea urchin g580 and sponge glyconectin 1 are promoting cell adhesion of hatched blastula ectodermal cells only in the presence of physiological calcium ion concentration. These results indicated that the molecular mechanism of highly polyvalent GN1 epitopes may represent the basis of attachment of ectoderm to apical lamina glycocalyx (Figure 2, Figure 3 and Figure 4).

#### 2.4.2. Inhibition of Blastula Cell Reaggregation with the Anti-GN1 Block 2 Monoclonal Antibody

Monovalent Fab fragments of the anti-GN1 Block 2 monoclonal antibodies were added to 100 μL of 10^5^ dissociated hatched blastula cells at 150 μg/mL FSW. The inhibition of aggregation was scored after 1 h of slow rotation at 16 °C. As shown in Figure 5D, the Block 2 Fab fragments inhibited cell adhesion indicating the involvement of the endogenous glycan epitopes in the aggregation of sea urchin hatched blastula cells when compared with not treated culture (Figure 5C). The unrelated mouse Fabs also did not have effect. The specificity of inhibition was further confirmed as follows. Protein-free adhesion glycans isolated of sponge glyconectin 1 were preincubated with Fab fragments of the Block 2 monoclonal antibody. This glycan preparation abrogated the inhibition effect of the Block 2 Fab fragments when used in two-fold molar excess showing a similar aggregation score as in Figure 5C. As expected, the mixture of glycosaminoglycans (chondroitin sulfate A, B, and C, keratan sulfate, heparin, heparan sulfate, and hyaluronic acid at 700 μg/mL concentration) did not abolish antibody mediated inhibition of cell adhesion (not shown, the same absence of adhesion as in Figure 5D). Thus, the Block 2 Fabs prevented sea urchin cell reaggregation through specific binding the GN1 adhesion glycan epitope. The finding that hatched blastula cell adhesion could be promoted by glyconectin type of acidic glycans bearing the GN1 epitope (Figure 5A,B), and the degree of inhibition of cell adhesion by the Fab fragments of the anti-GN1 Block 2 monoclonal antibodies is selective and concentration dependent, showed that the GN1 epitope in g580 glycan is a cell adhesion promoting structure.

### 2.5. Adhesion of g580 Glyconectin Glycan Coated Beads

Since the GN1 epitopes of the sponge glyconectin 1 g200 glycan functions as cell adhesion molecule via highly polyvalent and calcium ion dependent glycan–glycan binding, and since sea urchin g580 immuno-purified glycans also contain the same highly polyvalent GN1 epitope which promotes sea urchin hatched blastula cell reaggregation, it was necessary to examine whether g580 glycans can also self-associate via GN1 self-interactions. In order to test directly the involvement of such molecular mechanism of glycan–glycan interactions, bead adhesion assay using g580 glycans adsorbed to latex-amidine beads was performed (see details under Material and Methods). The beads diameter of 0.6 μm was able to accommodate about 10 to 20 immunopurified and protein-free g580 glycan molecules. Bead aggregation was performed with 100 μL of 2% bead suspension in the presence of physiological calcium ions, (SWT—buffered with 10 mM Tris pH 7.2), as well as in the absence calcium ions (CMF-ASW—buffered with 10 mM Tris pH 7.2) that was used as a control experiment. The results presented in Figure 6A showed that without calcium ions present in sea water g580 beads did not aggregate, whereas in the presence of a physiological concentration of calcium ions in sea water g580 coated beads strongly aggregated as shown in Figure 6B, thus indicating glycan–glycan binding as the molecular mechanism of beads adhesion.

In order to show that the adhesion forces responsible for beads aggregation is driven by the GN1 epitope intermolecular self-binding between g580, Fab monovalent fragments of the anti-GN1 Block 2 monoclonal antibodies, selectively recognizing only the GN1 epitope, were used to inhibit such bead aggregation. Indeed, as shown in Figure 6C, in the presence of physiological calcium ion concentration in sea water 100 μg/mL of Fabs of Block 2 inhibited g580 bead adhesion, whereas the unrelated mouse Fabs did not have effect. Control beads without g580 coating did not aggregate in sea water under the same conditions (Figure 6D). These results indicated that calcium dependent glycan–glycan binding of the polyvalent GN1 epitope in g580 has sufficient binding strength to drive g580 coated bead adhesion.

Taking all reported data, it could be proposed that a similar molecular mechanism of glycan–glycan binding involving self-associations via polyvalent GN1 epitopes may represent the basis of ectoderm adhesion to apical lamina glycocalyx that enables blastulation and provides the sufficient adhesion forces necessary for keeping blastula and blastocoel embryonal structure intact (Figure 2, Figure 3 and Figure 4).

### 2.6. Alteration of Hatched Blastula Embryos Shape and Inhibition of Gastrulation by the Anti-GN1 Block 2 Monoclonal Antibody

The final in vivo experiments were performed in order to test the effect of monovalent Fab fragments of the anti-GN1 Block 2 monoclonal antibody on sea urchin hatched blastula embryo integrity and further morphogenesis following blastulation. Four different concentrations of the Block 2 monoclonal antibody Fabs were added to the culture medium as described under Material and Methods. As shown in Figure 7A, 10 µg of Fabs per 100 μL FSW containing 100 embryos have affected the integrity of hatched blastula by causing a change from smooth to the grape-like morula structure. This antibody induced dissociation of ectoderm from the apical lamina through loss of its gel-like structure but did not visibly affect desmosome mediated lateral cell–cell adhesion. Embryos treated with the saturating concentration of the Block 2 Fab fragments did not recover their original cell shape and did not develop further to the gastrula stage. Lower concentrations of the Block 2 Fabs, 5 µg, 2.5 µg, and 1.25 µg per 100 µL FSW containing 100 embryos did not have a visible effect on embryo shape but did delay gastrulation 8 to 12 h, that was proportional to the amount of the antibody added (Figure 7B–D). As in the control untreated culture (Figure 7F), embryos incubated with 10 µg of Fab fragments of the control C-16 monoclonal antibody (which recognized a different carbohydrate epitope) per 100 μL FSW containing 100 embryos did not induce any abnormalities in shape and delay in development (Figure 7E). These results, together with: (a) immuno-localization of the GN1 glycan epitope in the apical lamina of hatched *L. pictus* embryos, (b) GN1 involvement in reaggregation of blastula ectodermal cells, and (c) self-association of immunopurified g580 in bead adhesion assay, indicate that the GN1 glycan epitope is providing sufficient force for anchoring of the apical part of ectoderm cells via microvilli to the apical lamina matrix formed by g580 glycan–glycan self-associations (Figure 2 and Figure 4). This functional role of GN1 in the first morphogenetic step of blastulation is important because at hatched blastula developmental stage that is transitioning from morula to gastrula basal lamina is not yet completely formed. Thus, emergence of the blastula form, stabilization of blastocoel cavity, and embryo integrity at hatched blastula stage are all mainly dependent on the apical and lateral adhesion of ectodermal cells that is mediated by the GN1 epitopes and desmosome cadherins, respectively.

### 2.7. Model and Mechanism of Blastulation and Integrity of Hatched Sea Urchin Blastula on the Basis of Glycan–Glycan Interactions via the GN1 Epitope of g580 Glyconectin Glycan

In order to study the role of the GN1 epitope of the g580 glyconectin glycan in blastulation hatched blastula was chosen as the most appropriate since it is free of fertilization membrane and still does not have completely formed the basal lamina structure. Based on the presented biochemical and ultrastructural results a schematic drawing model of hatched blastula cells and their microvilli with associated glycocalyx via highly polyvalent glycan–glycan binding of GM1 epitope is constructed (Figure 2 and Figure 4). Green lines represent g580 carrying the GN1 adhesion glycan epitope in the glycocalyx space and keeping the structure of blastula through intermolecular self-associations of over 200 GN1 epitopes per one g580 glycan molecule. The amount of g580 and GN1 in the apical lamina of one cell was calculated on the basis of the recovered quantity of g580 and immunobinding of the anti-GN1. These measurements indicated that one cell is surrounded in the apical lamina part with at least 1000 and maximally to 10,000 g580 molecules. That is about 10^5^ to 10^6^ GN1 epitopes per cell. According to the atomic force microscopy measurements of the intermolecular binding strength measurements between GN1 epitopes under physiological conditions, 2 × 10^3^ GN1 repeats per one molecule provide binding strength of 0.4 nanonewtons (corresponds to binding strength between single pair of sponge glyconectin 1 molecules) [69]. Translating to the cellular level, that force would be sufficient to keep support the weight of 1600 in sea water. Therefore, the total force calculated for 2 × 10^5^ g580 (1000 g580 molecules × 200 GN1 epitopes/g580) epitopes present per single sea urchin hatched blastula cell would be 40 nanonewtons. Such an extreme, but gel-like elastic structure of glycocalyx provided by g580 glycan–glycan interactions could support to keep the weight of 160,000 cells in sea water and provide the mechanical basis for the formation of blastula shape and its maintenance, prior to the basal lamina formation during gastrulation, under natural environmental movements of sea water.

## 3. Discussion

Sea urchins hatched blastula has a simple round hollow form comprised of ectodermal cells. At this early developmental stage, embryos have released fertilization membrane but did not yet complete formation of the basal lamina. Therefore, intermolecular binding of the apical lamina glycocalyx matrix and lateral desmosome cadherins seem to be the major sources of forces for keeping integrity of the natural hatched blastula shape, whereas the additional interactions to the basal lamina via cell adhesion molecules such as fibronectin, laminin, collagen and other extracellular matrix glycoproteins, and integrins may be somewhat less significant [5,11,12,15,16,17,18,19,20,21,22,23,28,29,30,31,32,33,34,35,36,37,38,39,40,41,42,43,44,45,46,47,48,49,50,51,52,53,54,55,56]. In order to examine the possible cell adhesive role of previously reported glyconectin glycan g580 in blastulation related to the integrity of hatched blastula, transparent *L. pictus* embryos were used for in vivo and in vitro immune-histological and biochemical analyses. Since these embryos can be also easily obtained in large amounts and cultured in an inexpensive and simple manner, they permitted isolation of analyzable amounts of g580.

Ultrastructural electron microscopy revealed that *L. pictus* apical lamina extracellular matrix is aligning with the shape of the apical cell surfaces but does not intrude laterally between the cells. This matrix contains semitransparent material and dense fibers that both seem to connect it to the cell membrane and membrane microvilli extending through it [28,57,58,59]. This is exactly the same spatial localization of the g580 acidic glycan carrying the GN1 epitope in the hatched blastula stage that was shown here by confocal and electron microscopy. Furthermore, presented results of in vivo experiments using saturating concentration of monovalent Fab fragments of the anti-GN1 Block monoclonal antibody revealed that without binding forces, provided by calcium ion dependent GN1 adhesion epitope interactions of g580 acidic glycan, the normal blastula shape was lost and changed to a morula-like appearance. Consequently, further development towards gastrula was blocked and embryos were arrested at this stage. Since the g580 acidic glycan, which contains the GN1 epitope, is one of the major glycocalyx components of the apical lamina matrix where microvilli are embedded, it seems that g580 interactions are the major source of forces for keeping normal hatched blastula shape prior to the complete basal lamina formation. The results obtained with the lower non-saturating concentrations of the Block 2 Fabs, that did not cause visible embryonal form changes, could be interpreted at the molecular level as the incomplete blocking of the highly polyvalent GN1 interactions that were still capable of maintaining the robust gel-like structures of the apical lamina. The concentration dependent delaying effect of non-saturating concentrations of antibodies on the transition towards gastrula, when compared with untreated embryos and control antibodies treated embryos, indicated that the incomplete assembled apical lamina and the partial blocking of the GN1 adhesion sites that are serving as anchoring points may also be involved during movement of primary and secondary mesenchymal cells into the blastocoel. Furthermore, the additional in vitro reaggregation experiments have confirmed that g580 bearing the GN1 epitope is necessary for cellular adhesion. Together with the calcium ions dependent g580 coated bead adhesion via GN1 interactions these results indicated that the essential molecular mechanism driving cell adhesion is based on g580 glycan–glycan binding via GN1 epitopes schematically presented in Figure 2 and Figure 4. However, the nature of the core protein to which g580 glycan is covalently linked, and the ratio between the transmembrane and secreted forms are still unknown.

Here presented in vivo experiments with the anti-GN1 monoclonal antibodies causing a reversal of smooth hatched blastula shape to a morula-like form indicated that desmosome cadherins, that are mediating the lateral adhesion of ectodermal cell, are not sufficient for maintaining the normal blastula shape. Furthermore, incubation of hatched blastula embryos in calcium and magnesium ion free sea water resulted in a morula-like shape that also caused the unbinding between g580 and the loss of the apical lamina glycocalyx gel-like structure. These results implied that g580, the major acidic glycan components of the apical lamina glycocalyx in hatched blastula, is one of the essential molecules establishing and maintaining native glycocalyx structure and embryo integrity.

Based on a large amount of published data and results presented here it can be concluded that the anatomical integrity of sea urchin embryos at the hatched blastula stage is based on lateral, apical, and basal cell adhesion. Most of the studies in this field, presented in the introduction section, are focused on the process of gastrulation and less on earlier blastulation. At the gastrula stage major morphogenetic movements resulting in the ectoderm, mesoderm, and endoderm layers formation are tightly connected to the basal lamina construction and cell to cell and cell to basal lamina adhesion. At the blastula stage, as discussed above, preservation of an embryo structure is more dependent on the apical lamina matrix structure and microvilli to apical lamina glycocalyx adhesion, and not on the building of higher pressure in the blastocoel. This was previously shown not to be possible [9]. Hyaline and echinonectin, together with here reported glyconectin g580 acidic glycan, that is carrying GN1 adhesion epitope, are the major components of the apical lamina glycocalyx matrix that are keeping the embryo shape and integrity [12,15,16,40,41,42,43,44,45,46,47,48,49,50,51]. At the stage of hatched blastula that is transitioning to the beginning gastrula, it was reported that apical cell adhesion via echinonectin and hyalin decreases [42]. In order to keep the normal and functional blastula embryo integrity through such changes in intermolecular interactions, the remaining strong but reversible adhesion may be provided via highly polyvalent Velcro-like glycan–glycan interactions of g580 GN1 epitopes. The apical lamina anatomical shape reassembles the gel hollow sphere established in part by glycocalyx GN1 glyconectins. This would provide anchoring points for the apical part of ectodermal cells via GN1-GN1 binding. According to the previous biophysical and biochemical studies at the molecular and cellular level GN1 self-associations would be able to sustain robust but reversible binding that is providing soft gel-like flexibility [1,72,73,74,75]. Here reported evidence obtained by in vitro and in vivo experiments, in accordance with the above discussed published reports, indicate: (a) that multiple intermolecular binding between these three types of molecules results in the assembly of the apical lamina glycocalyx matrix, and (b) that these molecules are involved in association with the plasma membranes, including microvilli membranes. Such orchestrated multimolecular assembles seem to be essential for the maintenance of stabile but relatively flexible blastula form. However, it should be noted that the exact molecular mechanisms of binding, as well as quantitative measurements of intermolecular forces are not completely evaluated for most of these molecules, with the exception of the GN1 glycan structure.

Several groups have repeated the original atomic force measurements of the binding strength for the natural sponge glyconectin 1, the purified sponge g200 glyconectin glycan, the pure synthetic GN1 epitope, and the control GN1 epitope structure without sulfate [77,78,79,80]. All obtained results confirmed the specificity of the polyvalent glycan–glycan binding via self-association of the GN1 epitope structures. Since glyconectin glycans have such extreme binding selectivity between specific epitope structures they could also provide the function as the first biosensors interacting with non-self-environment [69,77,78,79,80,81,82].

It was astonishing that the calculated 40 nanonewtons force could be provided by GN1 self-binding between g580 molecules present in glycocalyx surrounding the apical part of a single ectodermal cell. This single self-binding strength can keep the weight of 160,000 cells and could certainly allow apical lamina glycocalyx gel-like structure as adhesion anchoring extracellular space for the apical microvilli cell surfaces. Therefore, environmental sea wave forces would not be able to destroy the integrity of embryos. Furthermore, the gel-like structure of g580 glycocalyx can absorb sharp shocks and prevent the entry of foreign particles, but enables selective diffusion of small molecules, thus displaying some similarities to other more complex cell–cell and cell–matrix attachments structures observed in the blood brain barrier [83] or in neuronal networks [84], amphibian eggs [85,86], and in mucosal membranes [87].

It should be stressed that the adhesion forces provided by glycoprotein and glycolipid interactions located in the plasma membrane and extracellular matrices such as apical glycocalyx, basal lamina, and, also, less structured assemblies in tissues, are providing the physical force that maintains the shape and the anatomical integrity of multicellular organisms [67,68,69,81,82]. Molecular interactions are mostly determined by employing various kinetic measurements that are providing affinity binding constants, less by calorimetric measurements that are resulting in binding energies, and least in the actual direct measurements of binding forces by atomic force microscopy, as conducted with the GN1 epitopes present in glyconectins [69,77,78,79,80,81,82]. Indeed, atomic force microscopy measurements of GN1 polyvalent glycan–glycan binding strength have directly connected the structure of this epitope to its cell adhesion function. The obtained results presented here allow the conclusion that intermolecular binding force driving self-assembly of g580 in glycocalyx via GN1 epitopes can support the mechanical stability of the hatched blastula.

Research on glyconectin type of glycan is still missing the complete structural analyses due to their: (a) high molecular mass, usually over 10^5^ D, (b) more complex structures when compared to classical glycosaminoglycans heteropolymers of disaccharide repeated sequences, as well as to other homopolymers of glycans, and (c) the lack of structure specific glycosidases. Although in the absence of the complete sequencing of g580 by nuclear magnetic resonance and mass spectrometry here reported biochemical and immunochemical data obtained by binding and blocking experiments of the Block 2 monoclonal antibody to purified g200 and g580 glycans, to the natural and synthetic GN1 epitope in sulfated and de-sulphated form, to all known glycosaminoglycans structures, to mammalian glycoproteins carrying classical *N-* and *O-linked* glycans, to fucose and 3 sulfated N-acetylglucosamine monosaccharides, showed that the anti-GN1 Block 2 monoclonal antibody specifically recognize only GN1 sulfated disaccharide epitope and thus g580 indeed carries GN1 epitope structure. Furthermore, isolated and immunopurified g580 were not degraded with any known glycosaminoglycan degrading enzymes, such as hyaluronidases, chondroitinases, and heparinases, or endo- or exo-glycosidases commonly used for *N-* and *O-linked* glycan degradation [69,70,71,72,73,74,75,76,77,78,79,80,81]. These evidence together with the previous partial structural and sequence analyses of g200, imply that glyconectin type of glycans that are found in sponges, sea urchin embryos, and human colon carcinomas glycocalyx, are mainly linear complex heteropolymers that contain repeats of specific types of self-adhesion structures such as GN1. In nine different species of sponges, 16 glyconectin adhesive epitope structures were partially sequenced. Several of them were shown to mediate species-specific cell adhesion that was responsible for self-recognition enabled through highly polyvalent glycan to glycan interactions [1,70]. Although some of the adhesive epitope structures, such as GN1, are shared with other evolutionary more advanced species, the complete structure of glyconectins is species-specific. The existence of such interspecies sharing of same glycan epitope structures, as well as species-specificity of other glycan structures, were demonstrated with numerous examples that were extensively reviewed [88,89]. However, this observation is not yet possible to explain due to the lack of more complete analyses of the enormously heterogeneous glycan structures in the variety of species.

As discussed in more detail above, it should be noted that glyconectins have specific structural characteristics that are reflected in their unique monosaccharide composition and physicochemical properties. These features, different from other glycans, may have been the cause of their invisibility when using the common biochemical and microscopical study approaches [1]. Therefore, one of the important future research perspectives should be related to isolation and sequencing of glycosyltransferases, sulfotransferases, and glycosidases specific for GN1 and other glyconectin adhesive structures biosynthesis and degradation. Regulation of expression of these enzymes during early development of sea urchin, as well as in sponges and human cancer tissue, would contribute to more advanced knowledge of indirect coding and regulation of biosynthesis of specific glycan structures via glycosyltransferases, as well as to easier complete sequencing of their structures by specific glycosidases. These perspective studies on glyconectin glycan–glycan mediated adhesion related to morphogenesis and self-recognition would advance the knowledge about their structure and function at the molecular and cellular level.

From the vastly acquired data in the broad field of cell adhesion related to many biological functions associated with morphogenesis, maintenance and regeneration processes in adult organism life, self-recognition and immune response, and in pathological cases such as cancer growth and metastasis, it can be concluded that many different classes of glycoproteins and glycolipids with precise spatial and temporal expression patterns in the plasma membrane and extracellular matrices are operating via coordinated and multistep protein to protein, protein to glycan and glycan–glycan interactions. These three fundamental types of intermolecular binding can have different affinities and, respectively, different binding strength. Research on glyconectins, the relatively new class of large acidic glycans, is still slowly emerging with few studies that are showing their relevance in cell adhesion as well as in cell recognition. The importance of these interactions is based on often overlooked facts that glyconectin glycans, besides glycosaminoglycans on proteoglycans, and small glycans on mucin, are the most abundant dense layer of molecules in the outermost glycocalyx of cell surfaces. Since glycans cannot be physically avoided as the initial encounters of the cell environment they must be involved in the first steps of cellular adhesion.

## 4. Materials and Methods

### 4.1. Fertilization and Embryo Cultures

*Letychinu pictus* specimens were obtained from Marinus Inc. CA. Sperm and eggs were released after intracelomic injection of 0.5–1 mL of 0.5 M KCl [90]. Eggs were passed through 253 µm nylon mesh and were washed three times with natural filtered sea water-(FSW) (0.22 µm Millipore was used for filtration) by 1× *g* sedimentation. About 2% *v/v* egg suspension per 100 mL of FSW were fertilized with 10 drops of freshly diluted sperm (1:20). After three washing of fertilized eggs by 1× *g* sedimentation, 1–2 × 10^4^ embryos/mL FSW were cultured in the presence of 50 mg gentamycin or 50 mg/mL penicillin and streptomycin using a 10 rpm rocking shaker at 16 °C.

### 4.2. Dissociation of Embryos

At the blastula stage embryos were washed two times with FSW and once with Ca^2+^ and Mg^2+^ free filtered sea water buffered with 0.18 g/L of NaHCO_3_ (CMF-ASW) by centrifugation at 800 rpm. Dissociation in the CMF-ASW was proceeded by gentle pipetting of embryos with a polished glass pipette. The single cell suspension was washed two times in CMF-ASW by centrifugation at 1000× *g* for 1 min and finally resuspended in FSW.

### 4.3. Reaggregation of Dissociated Hatched Blastula Cells

100 µL of 10^5^ cells in FSW were placed in a small chamber prepared by sealing a rubber ring (1 cm in diameter and 0.2 cm high) with equal weights of warmed vaseline, lanolin, and mineral oil onto a glass coverslip. When testing inhibition or promotion of aggregation, 15 µg of either antibody, or g580, or sponge glyconectin 1 glycans at different concentrations were added to cells. Incubations were carried out in a moist chamber at 16 under gentle rotation and rocking at 10 rpm or by gentle mixing of the cells every 2 min.

### 4.4. In Vivo Treatment of Hatched Blastula Embryos with the Anti-GN1 Fab Fragments of the Block 2 Monoclonal Antibody

The Block 2 anti-GN1 monoclonal antibody was developed by Dr. Gradimir Misevic and is not yet commercially available. Procedures describing development and testing of the specificity of this monoclonal antibody, as well as the preparation of monovalent Fab fragments were performed as previously described [70,71,72,73,74,75].

*L pictus* embryos at the hatched blastula were grown in 100 μL FSW, using the same chamber as described under the previous reaggregation method section. Different concentrations of the anti-GN1 Fab Fragments of the Block 2 monoclonal antibody or the control C-16 monoclonal antibody were added. Inhibition of gastrulation and shape changes were continuously scored for 24 h of incubation at 16 °C.

### 4.5. Preparation and Aggregation of g580 Coated Beads

Latex-amidine beads (Molecular Probes) with a diameter of 0.6 μm were suspended in 200 μL of 10 mM Tris pH 7.4 buffered CMF-ASW at a final concentration of 2%. Beads were washed three times by centrifugation at 3000× *g* for 10 min, resuspended in 200 μL of SWT, and sonicated for 5 min. 100 μg of either sea urchin g580 glycans or sponge glyconectin 1 were added to 200 μL of 2% bead suspension in 10 mM Tris pH 7.4 buffered CMF-ASW for 15 min at room temperature in order to allow adsorption. Unbound glycans were washed with the same incubation buffer by centrifugation at 3000× *g* for 10 min and resuspension in CMF-ASW. Aggregation of coated or untreated beads was performed in 100 μL of 2% suspension in either SWT or CMF-ASW, in the absence and presence of Fab fragments of the anti-GN1 Block 2 monoclonal antibodies.

### 4.6. Fixation and Embedding of Embryos

Embryos washed two times in FSW were resuspended and mixed with an equal volume of fixative (2% paraformaldehyde and 0.2% glutaraldehyde in 85% artificial sea water (ASW) buffered with 100 mM pH 7.4 cacodylate) for 1 h at room temperature. Embryos were washed two times with 85% ASW buffered with 50 mM cacodylate pH 7.4 and were subsequently incubated for 1 h in 0.1 M glycine in 85% ASW/cacodylate. After two additional washing in 85% ASW/cacodylate, embryos were dehydrated with ethanol and embedded in K4M.

### 4.7. Immunogold Staining of Embryos

For immunogold staining, ultra-thin sections of embryos were preincubated for 30 min at room temperature with 0.125 M NaCl, 0.1% Tween 20, 0.1% BSA, 0.5% fish gelatin, and 20 mM Tris pH 7.4. (TBF). After aspiration of TBF, 10 µL drop of the Block 2 monoclonal antibody in the same buffer solution at a final concentration of 100 µg/mL was added to the sections and incubated for 12 h at room temperature in a moist chamber. Sections were then washed five times with TBF, and then for 2 h in anti-mouse antibody conjugated to 30 nm gold particles (Auro probe). Sections were subsequently stained with uranyl acetate and lead citrate. Electron micrographs were taken on a Philips 3000 electron microscope.

### 4.8. Confocal Immunofluorescence Microscopy of the Whole Embryo Mounts

Embryos at the blastula stage were washed three times with FSW. Resuspended embryos were then fixed by the drop-wise addition of cold methanol (−20 °C). After stepwise rehydration, embryos were incubated with the Block 2 monoclonal antibody (100 µg/mL ASW containing 2% BSA) under gentle agitation at room temperature. After 2 h embryos were washed four times with ASW containing 1% BSA. FITC-labeled anti-mouse antibody (Jackson Laboratories Bar Harbor, ME, USA) at a concentration of 50 µg/mL ASW in 1% BSA was added, and incubation proceeded for 2 h at room temperature. Embryos were then washed four times with ASW containing 26 mg/mL diazobicyclo octane. Embryos were mounted between glass coverslips of 1 1/2 and 1/2 thickness using 1/2 coverslips as spacers. A Zeiss LSM confocal microscope and Vital images software were used for viewing embryos. Laser scans of fluorescence and transmitted light were taken at 0.6 µm steps using the Zeiss objective with 63× magnification and 1.4 numerical aperture.

### 4.9. Isolation of Glycans from Sea Urchin Embryos

Embryos were washed two times with filtered sea water (FSW) and subsequently delipidated [91]. The pellet was dried and suspended in 0.1 M Tris pH 8.0 containing 1 mM CaCl_2_. 10 mg/mL pronase dissolved freshly in the same buffer was added to the sample at a 1:20 dilution and incubated for 24 h at 60 °C. The same procedure was repeated three times. In order to remove amino acids and small peptides, the sample was applied to a G-25 Sephadex column and eluted with 50 mM pyridine acetate pH 5.0 [75]. The lyophilized glycan fraction was dissolved into 0.1 M Tris pH 8.0 containing 1 mM CaCl_2_ and 10 mM MgCl_2_ and was further treated with 1 mg/mL of DNAase I for 6 h at 37 °C. Subsequently, a new aliquot of pronase was added, and digestion was followed by gel filtration as previously described [75].

### 4.10. Immuno-Purification, Immunoblotting, and Polyacrylamide Gel Electrophoresis of Glycans

The excess of the Block 2 monoclonal antibody (1 mg/mL of 2% suspension of beads in 0.2 M NaCl and 20 mM Tris pH 7.4) was used for coating of carboxylated polystyrene beads (1 µm in diameter white polystyrene obtained from Molecular Probes). After 2 h incubation at room temperature beads were washed 3 times by 5 min centrifugation at 1000× *g*. Immuno-purification of glycans was performed in 0.2 M NaCl and 20 mM Tris pH 7.4 by incubation of 2% suspension of antibody coated beads with 1 mg/mL of total glycans isolated from blastula sea urchin embryos for 2 h at room temperature. After 5 min centrifugation at 1000× *g*, non-bound material was collected, and beads were washed two more times with 0.2 M NaCl and 20 mM Tris pH 7.4. Bound material was released from the beads after complete digestion of antibodies by overnight digestion with pronase (1 mg/mL) at 60 °C.

Gel electrophoresis and electroblotting were performed as described previously [72,92]. Immunodot assay was essentially performed as described previously [75], and [92]. Then, 0.5 µL of glycans of 1 µg/mL of water were pipetted onto a strip of DEAE cellulose paper (Schleicher and Schuell Munich, Germany). The Block 2 monoclonal antibody (100 µg/mL of 0.125 M NaCl, 1% BSA, and 20 mM Tris pH 7.4) was added to DEAE paper strips. After 2 h of shaking at room temperature, the paper was washed and incubated with anti-mouse peroxidase conjugated antibody (peroxidase-conjugated antibody to mouse immunoglobulins (DAKO-immunoglobulins Dakopatts, Denmark) for 2 h at room temperature. After washing the paper to remove the unbound antibodies, the color reaction was developed with chloronaphtol as described previously [75].

### 4.11. Analytical Methods

Neutral hexose was determined colorimetrically as described by [93], and uronic acid was assayed using the carbazol reaction [94]. Amino acid analysis was performed by the Pico-Tag method after hydrolysis of the samples in 6 M HCl vapor [95]. Monosaccharide composition of glycans was measured by gas chromatography combined with mass spectrometry as per-trimethylsilyl [96] and per-heptafluorobutyryl derivatives [97]. Methylation of was performed as described previously [98]

SO_4_^2−^ was determined by Dionex high pH anion-exchange chromatography, upon 12 h hydrolysis with 6 M HCl at 100 °C.

## 5. Conclusions

Presented biochemical and immuno-histological analyses showed that the immunopurified g580 is a glyconectin type of glycan, which expression is controlled in the strict spatio-temporal manner during embryogenesis of sea urchin *L. pictus* with the maximal presence in hatched blastula and gastrula. The g580 molecule is carrying about 200 repeats of β-d-Glc*p*NAc3S-(1→3)-α-l-Fuc*p* GN1 adhesion epitope which represents about 20% of its overall structure. The g580 glycan is one of the major acidic glycans in hatched blastula and gastrula stages.

Confocal and electron immune-microscopic analyses of hatched blastula with the Fab monovalent fragments of the Block 2 anti-GN1 monoclonal antibodies, revealed that the g580 glycan, carrying the highly polyvalent GN1 epitope, is expressed exclusively in the glycocalyx of the apical lamina associated with microvilli, besides Golgi complex where it is synthesized.

Functional in vitro cell adhesion experiments demonstrated that the sea urchin g580, as well as the sponge g200 glyconectins, can promote reaggregation of hatched blastula cells. Involvement of the GN1 epitope structure as the functional polyvalent unit in sea urchin g580 was established by the inhibition of the native reaggregation of dissociated cells via endogenous g580 with the monovalent Fab fragments of the anti-GN1 Block 2 monoclonal antibodies, and by the promotion of aggregation by both g580 and g200 glyconectin molecules.

The molecular mechanism of g580 involvement in cell adhesion was tested with g580 coated beads. The bead aggregation occurred only when they were coated with immunopurified and protein-free g580 glycan in the presence of physiological sea water calcium ion concentration, as in the case of the sponge g200. This aggregation was blocked by the Fab fragments of the anti-GN1 Block 2 monoclonal antibodies. Therefore, it could be concluded that glycan–glycan binding of g580 is the result of intermolecular associations of their polyvalent GN1 epitopes that are providing the main driving forces for bead, as well as cell adhesion of reaggregating hatched ectodermal cells.

In vivo treatment of hatched blastula embryos with Fab fragments of the anti-GN1 block 2 monoclonal antibodies caused the loss of blastula structure and inhibition of gastrulation. These results implied that the GN1 epitope structure of g580 glyconectin glycan is involved in blastulation and maintenance of blastula native structure after the release of the fertilization membrane and prior to the complete formation of the basal lamina.

On the basis of the previous measurements of: (a) glycan–glycan intermolecular binding strength between GN1 epitopes, (b) the high polyvalency of GN1 in g580, (c) the amount of g580 in the apical lamina glycocalyx and microvilli, and (d) the cell and bead adhesion involving the polyvalent interactions of GN1 epitopes carried by g580, it could be calculated that the 40 nanonewtons intermolecular associations of g580 GN1 epitopes can hold the weight of 160,000 cells in sea water. Therefore, GN1 binding can provide anchoring adhesion force of microvilli to the apical lamina and preserve its glycocalyx gel-like structure that is sufficient to establish, maintain, and preserve hatched blastula.

## Figures and Tables

**Figure 1 molecules-26-04012-f001:**
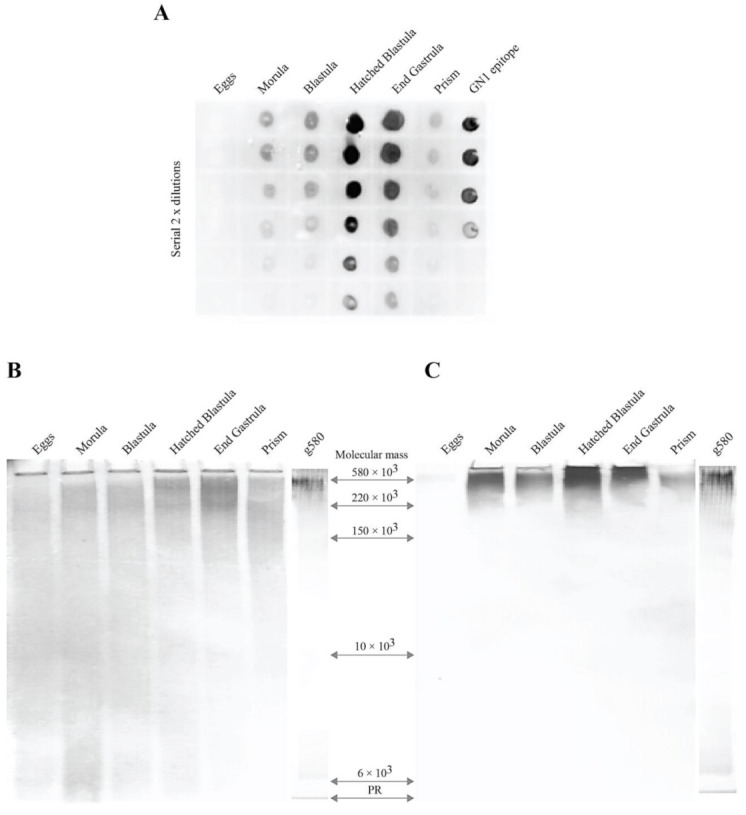
Expression of the GN1 epitope during *L. pictus* sea urchin development. (**A**) immunodot assay of total sea urchin glycans isolated from six developmental stages with the Block 2 monoclonal antibody. Then, 0.5 μL of glycans of two-fold dilutions starting with the concentration of 1 μg/μL water were spotted onto a DEAE-nitrocellulose paper. After brief drying DEAE-nitrocellulose was immunodecorated with the Block 2 monoclonal antibody and stained with anti-mouse peroxidase conjugated antibodies as described in Material and Methods. (**B**) polyacrylamide gel electrophoresis on 5–20% gradient gels of 25 μg sea urchin total protein-free glycans prepared from six different developmental stages, and 2.5 μg of immunopurified g580 sea urchin glycan from hatched blastula stage glycans were performed as described under Material and Methods. Gels were stained with Alcian blue for detecting acidic glycans. (**C**) immuno-detection of the GN1 epitope by the Block 2 monoclonal antibody in sea urchin glycans after gel electrophoresis as in (**A**), and electro-blotting. Glycosaminoglycans, heparin of M_r_ = 11 × 10^3^ D, chondroitin sulfate of M_r_ = 20 × 10^3^ and M_r_ = 100 × 10^3^ and hyaluronic acid of M_r_ = 225 × 10^3^ D, and sponge glyconectins g200 of M_r_ = 200 × 10^3^ D and g6 of M_r_ = 6 × 10^3^ were used as molecular weight standards.

**Figure 2 molecules-26-04012-f002:**
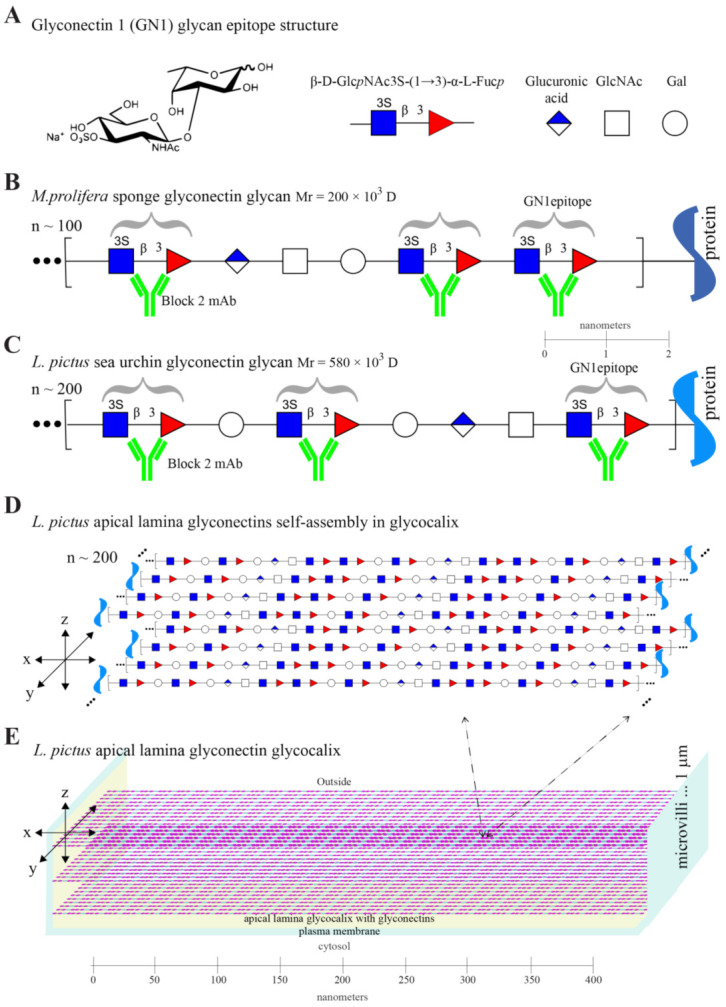
Structure of the GN1 cell adhesion epitope and molecular model of glycocalyx and part of microvilli. (**A**) Glyconectin 1 (GN1) glycan epitope structure. (**B**) *M. prolifera* sponge glyconectin glycan. (**C**) *L. pictus* sea urchin glyconectin glycan. (**D**) *L. pictus* apical lamina glyconectins self-assembly in glycocalyx. (**E**) *L. pictus* apical lamina glyconectin glycocalyx.

**Figure 3 molecules-26-04012-f003:**
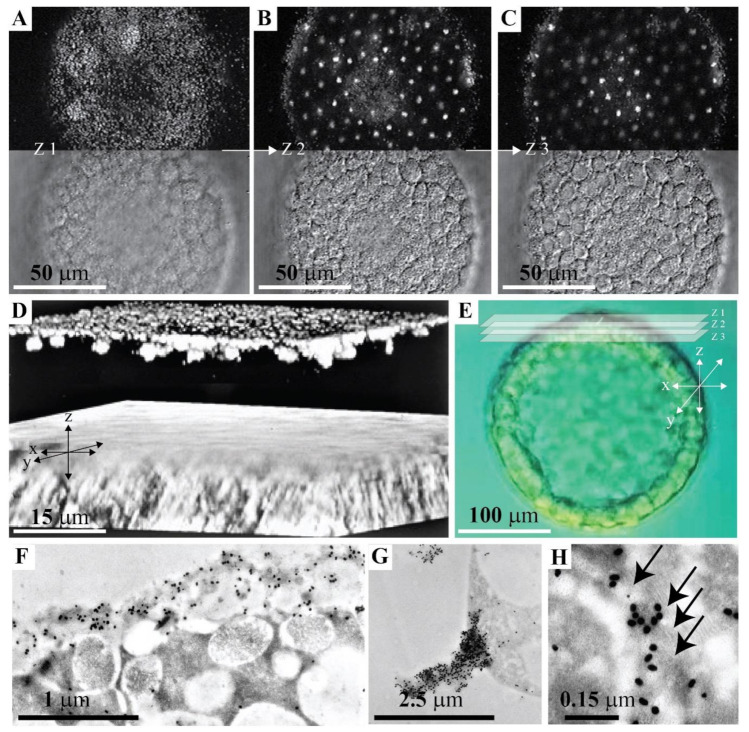
Immuno-histological localization of the GN1 glycan adhesion epitope in *L. pictus* hatched blastula. (**A**–**C**) immunofluorescence confocal microscopy of *L. pictus* hatched blastula stained with the ant-GN1 Block 2 monoclonal antibody. Whole mounts of blastula L. pictus embryos were stained with 100 μg/mL Block 2 as described in Materials and Methods. (**A**–**C**) parfocal series of compost images from fluorescence (upper image) and transmitted light (lower image) were taken with Zeiss 63× objective every 0.6 μm in *z*-axis starting from the apical part of ectodermal cells. Image in *z*-axis at distance from scan beginning (**A**) 0.0 μm (**B**) 1.2 μm, (**C**) 2.4 μm, (**D**) three-dimensional reconstruction of the embryo part obtained from images scanned with 63× objective, (**E**) image of the whole embryo, (**F**–**H**) immunogold electron microscopical localization of the glycan adhesion epitope in *L. pictus* hatched blastula, (**F**) apical lamina with microvilli, (**G**) part of microvilli, and (**H**) Golgi complex. Arrows mark Golgi.

**Figure 4 molecules-26-04012-f004:**
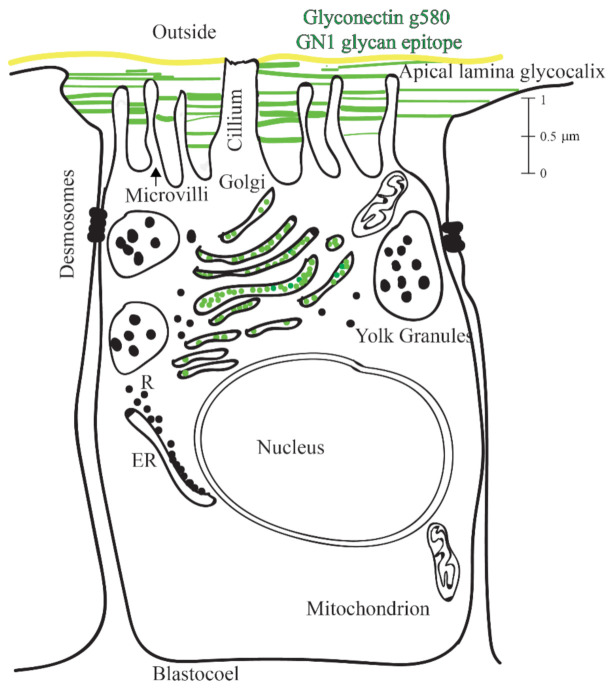
Model of hatched blastula apical lamina glycocalyx association with microvilli by the g580 glyconectin acidic glycan.

**Figure 5 molecules-26-04012-f005:**
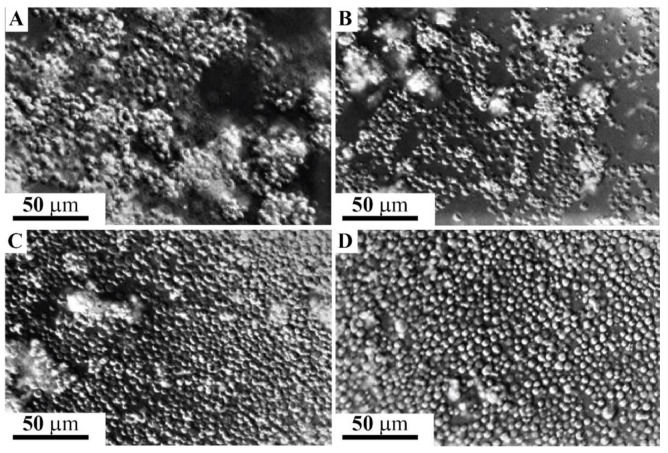
Hatched blastula cell reaggregation. (**A**) promotion of sea urchin hatched blastula cell aggregation by 15 μg of the g580 glycan in the volume of 100 μL FSW containing 10^5^ cells was performed using gentle rotation at 16 °C for 10 min, (**B**) aggregation promotion by 15 μg of the sponge g200 glycan under the same condition as in (**A**), (**C**) not treated culture of 105 dissociated hatched blastula cells, and (**D**) inhibition of sea urchin hatched blastula cell reaggregation by 15 μg of the anti-GN1 Fab fragments of the Block 2 monoclonal antibody was performed as in (**A**,**B**), but for 1 h.

**Figure 6 molecules-26-04012-f006:**
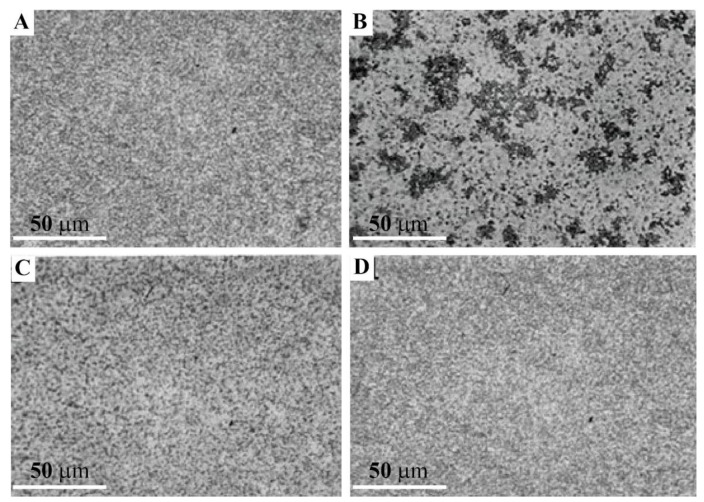
Intermolecular glycan–glycan binding of g580 glyconectin glycans demonstrated by adhesion of g580 coated beads. Aggregation of 100 μL 2% bead suspension: (**A**) g580 glyconectin glycans coated beads in the absence of physiological calcium ions (CMF-ASW—buffered with 10 mM Tris pH 7.2), (**B**) g580 glyconectin glycans coated beads in the presence of physiological calcium ions (SWT—buffered with 10 mM Tris pH 7.2), (**C**) g580 glyconectin glycans coated beads in the presence of physiological calcium ions (SWT—buffered with 10 mM Tris pH 7.2) with Fab monovalent fragments of the anti-GN1 Block 2 monoclonal antibodies (100 μg/mL), and (**D**) non coated beads in the presence of physiological calcium ions (SWT—buffered with 10 mM Tris pH 7.2).

**Figure 7 molecules-26-04012-f007:**
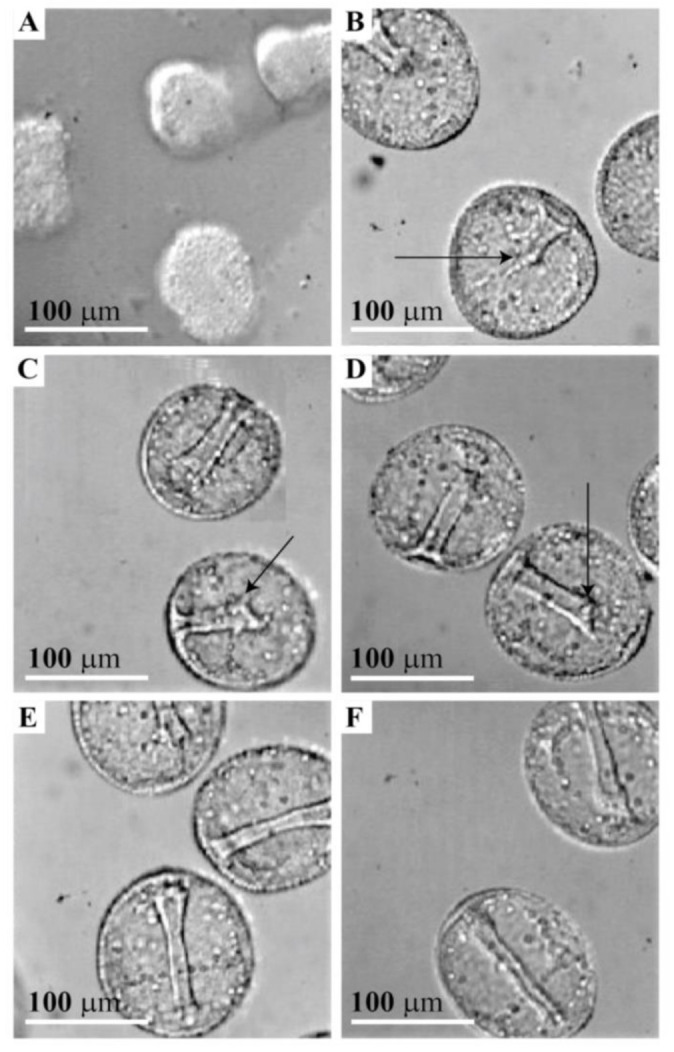
Alterations of embryo integrity and inhibition of gastrulation by the anti GN1 Block 2 monoclonal antibody. Hatched blastula embryos were grown in 100 μL FSW to which different concentrations of purified monoclonal antibodies were added. Inhibition of gastrulation and shape changes were scored during 24 h incubation at 16 °C with the (**A**) 10 μg Block 2, (**B**) 5 μg Block 2, (**C**) 2.5 μg Block 2, (**D**) 1.25 μg Block 2, (**E**) 10 μg control mouse C-16 monoclonal antibody, and (**F**) control untreated embryos. Arrows mark archenteron.

## Data Availability

Data is contained within the article.

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
