# Peer review of "Glyconectin Cell Adhesion Epitope, β-d-GlcpNAc3S-(1→3)-α-l-Fucp, Is Involved in Blastulation of Lytechinus pictus Sea Urchin Embryos"

_molecules, 2021, doi:10.3390/molecules26134012_

Round 1
Reviewer 1 Report
In this paper Misevic et. al. identified glyconectin adhesion epitope involved in blastulation of Sea Urchin embryos. It’s interesting and the very good science but, the manuscript writing style is not up to the mark. The sentences are very complicated and need to be simplified.
Here are the other shortcomings in the manuscript:
- Abstract: The authors’ know a lot about the topic but, they have written in a manner which is very confusing. There is lot of information in the abstract that is fit to be in an abstract of a review article but not in a research paper. Eg. There are no experiments in the paper that measured 40 nanonewton force that holds 16,000 cells under water. That work was done by other groups and it could be in the introduction or discussion but not abstract. Whole abstract should be rewritten in a manner that it mentions ONLY the data tested in this paper.
- Introduction/Discussion: The sentences are complicated and the narrative keeps meandering between Porifera and cancer. My suggestion is to dedicate a paragraph to it in the introduction and then again in the discussion. I don’t think it is a good idea to mention it in the abstract as there are no comparative experiments or data presented in the manuscript. In fact, it is mentioned so many times throughout the manuscript but no comparative data or diagram was presented. Why not add one as that would increase the impact of the information.
- Materials/Methods: Whole section needs to be rewritten as it is missing out critical information. Eg: a) Where did you get the GN1/Block antibodies? Please mention company and product code. Same with the latex amidine beads (product code). What is LSM microscope’s model number? What was the iteration number, gain, excitation wavelength/ emission range etc.? What software did you use to analyze data? Lens oil (specifics)…. FITC-antibodies…same thing… product code? These are just few examples. Please be proactive and revisit each section as each of them is missing critical information required to reproduce the data. Please write in excruciating detail. This paper cannot get published if you do not provide adequate information for each experiment.
- Results:
- Fig 1. Overall: If this experiment was done once then this is a very unreliable experiment and should not be in the manuscript. If you have repeated it few times, then please provide quantification of the bands and provide a statistical analysis.
- Fig 1. Specifics: Did you try running it in a handmade 4% gel only? Did you run your protein devoid of glycans ever? I believe that is an important control your gel is missing. That would be a necessary control for the immunoblots too. That would test and prove the specificity of your antibodies as well.
- Fig 1a: A quantitate analysis along with statistics is critical for this data.
- Fig 1c: The Immunoblot is overexposed after looking at the whole membrane. It could also be a problem of not blocking the membrane properly. Either way needs to be replaced or redone.
Specifics about anti-mouse peroxidase in the material/methods needs mention.
- Fig 3. Critical Details in materials/ methods missing. How do you know it’s Golgi/microvilli? Where is your co-localization control? Extremely critical controls missing hence the conclusion is unreliable and it’s a mere assumption.
- Fig 5. A quantitative analysis of this data is required. Section 2.4.1 is leaving out some critical information as “data not shown”. Please add that as I think those are important. Please add them as supplements.
- Fig 6. The figure legend and the figure is highly misleading. 6B Looks less aggregated compared to 6A. What is the difference between 6A and 6C? It’s not evident and very confusing even after reading the results section.
- Fig 7. The figure needs labeling. What are we looking at? Please use arrows to indicate what we are looking at.
- Last but not the least: The citations are so distracting and is breaking the narrative. If your citation is continuous then you write like this: eg citations mentioned in line 523, 524,525 will be written as: [5], [11], [12], [15-23], [28-56].
Please review the comments and paper carefully and add all the information requested. I refrained from pointing out each one of the missing experimental details but I would request you to review thoroughly and add those. Please add the data under (data not shown) section under supplementary section. Again, this is very good science and I would like to read the revised manuscript.
Reviewer 2 Report
The current manuscript entitled “Glyconection cell adhesion epitope, D-GlcpNAc3S beta1-3 L-Fucp, is involved in blastulation of Lytechinus pictus sea urchin embryos (Misevic et al., 1181527)” describes significance of glycan-glycan interaction in maintenance and preservation of blastula form in sea urchin eggs. The authors utilized anti-NS1 antibody for detecting intra- and extra-cellular distribution of a carbohydrate epitope (Figure 3), and demonstrated that the antibody greatly suppressed gastrulation (Figure 7) and reaggregation of dispersed blastula cells (Figure 5). These results propose significance of glycan-glycan interaction in these events.
Although glycan-glycan interaction is definitively one of the fundamental molecular interactions in biological processes, it has been difficult to prove biological significance of the interaction because of lacking of good methods and tools to achieve it. In this sense, the authors’ strategy to utilize the anti-GN1 antibody is unique and highly evaluated. Especially effect of the antibody on the process of gastrulation is quite interesting.
Major comments:
1) The glycocalyx mainly distributes just attached to the microvilli on the surface of the apical part of ectodermal cells in blastula form. If the dispersed ectodermal cells maintain cell polarity in suspension, it is reasonable that the two ectodermal cells attached each other along their own apical surface together. It is just the attachment but not likely to support the integrity of the blastula form. If this kind of glycocalyx is involved in the integrity of lateral, apical, basal structure, by what mechanism the apical interaction can support the integrity blastula form? Discussion on this point is recommended to add in the discussion.
2) Figure 5,6,7. Potential biological roles of the glycan epitope NS1 are suggested by the antibody-inhibition tests in the Figures. These experiments should require negative control, i.e., incubation with control (= nonspecific) antibody has no effect. This control experiment may be included in the figure or just commented in the text.
Minor comments:
1) Figure 2. The shown self assembly model is interesting, however evidence to support the model seems to be lacking in the text. It is likely that background evidence is shown in their previous data, therefore this figure is recommend to be commented as a background study in “Introduction" part. Blue squares and red triangles stand for D-GlcNAc3Sbeta1-3 and L-Fucp respectively, whereas three other marks (circles, squares and combined triangles) have not been defined.
3) In the present manuscript, “Results” part includes not only experimental results but other previous results (i.e., Figure 2) and discussions (i.e., 2.7 Model and mechanism of..). These statements should be included in “Introduction” and “ Discussion” respectively. The result part should include only experimental results and minimal explanation of materials and methods.
Reviewer 3 Report
The authors describe a very nice piece of work and this study strongly deserves to be reported after minor corrections.
1) please correct the nomenclature of GN1, following carefully IUPAC rules.
2) please indicate clearly the specificity for carbohydrates of the antibodies. Is for example Block 2 100% specific for GN1?
3) in the absence of NMR or MS spectra, how can the authors fully conclude that GN1 is the epitope in g580?
4) O- and N-carbohydrates should be in italics.
Author Response
Manuscript (molecules-1181527) entitled “Glyconectin Cell Adhesion Epitope, D-GlcpNAc3S β 1–3 L-Fucp, is Involved in Blastulation of Lytechinus pictus Sea Urchin Embryos”.
We have followed suggestions for amendments and corrections raised by reviewer 3 and appropriately revised our manuscript. Added text is in blue fonts and is marked with yellow.
We have also used now only institutional e mail addresses for all authors Page 1. Please note that opopescu.ubbcluj@gmail.com is the institutional address for Prof. Octavian Popescu at Institute for Interdisciplinary Research in Bio-Nano-Sciences, Molecular Biology Center, Babeș-Bolyai-University, 400084 Cluj-Napoca, Romania;
Reviewer 3 questions:
1) please correct the nomenclature of GN1, following carefully IUPAC rules.
We have corrected GN1 nomenclature now by using suggested IUPAC rules on page 1 line 3 and 20, page 3 line 122 and 148, page 4 lines 157, 160, 172, 195, 198 and 199, page 5 line 223, page 8 line 290, page 9 line 328, page 11 line 389, and page 22 line 846.
Also, the text in Figure 2 for GN1 epitope sequence was amended by using IUPAC nomenclature.
2) please indicate clearly the specificity for carbohydrates of the antibodies. Is for example Block 2 100% specific for GN1?
Answers: We have amended text on page 4 lines 186 -194 by providing now also all additional evidence and explanations that Block 2 100% recognize GN1. The added text is: “Furthermore, the results presented here together with previously published data showed that purified natural de-sulfated disaccharides, as well as synthetically obtained sulfated and non-sulfated disaccharides, L – fucose, D - N-acetylglucosamine 3 sulfate, N- and O-linked glycans from mammalian, sponge and sea urchin glycoproteins, did not bind to the Block 2 monoclonal antibody and were not capable of inhibiting binding of this an-tibody to the GN1 epitope of g200 glycan. Contrary only purified and/or synthetically obtained GN1 sulfated disaccharide epitope structure bound to the Block 2 monoclonal antibody and was capable of inhibiting binding of this antibody to g200 [69] [70] [71] [72] [73] [74] [75] [76] [77] [78] [79] [80] [81] [82]. These findings revealed strict specificity of the Block 2 monoclonal antibody to the GN1 epitope sequence D- β-D-GlcpNAc3S-(1→3)-α-L-Fucp schematically presented in Figure 2 A-C [72] [73] [77] [78] [79].”
Page 8 line 299 added text is: “degraded by glycosidases specific for any known glycosaminoglycan structures”.
Also, we have extended references that provided the evidence for Block 2 specificity.
3) in the absence of NMR or MS spectra, how can the authors fully conclude that GN1 is the epitope in g580?
Answers: We have amended on page 18 lines 667 - 680. The added explanation with experimental evidence now fully allow the conclusion that the GN1 epitope is present in g580. The added text is: “Although in the absence of the complete sequencing of g580 by nuclear magnetic reso-nance and mass spectrometry here reported biochemical and immunochemical data ob-tained by binding and blocking experiments of the Block 2 monoclonal antibody to puri-fied g200 and g580 glycans, to the natural and synthetic GN1 epitope in sulfated and de-sulphated form, to all known glycosaminoglycans structures, to mammalian glyco-proteins carrying classical N- and O-linked glycans, to fucose and 3 sulfated N-acetylglucosamine monosaccharides, showed that the anti-GN1 Block 2 monoclonal antibody specifically recognize only GN1 sulfated disaccharide epitope and thus g580 indeed carries GN1 epitope structure. Furthermore, isolated and immunopurified g580 were not degraded with any known glycosaminoglycan degrading enzymes, such as hy-aluronidases, chondroitinases, and heparinases, or endo- or exo-glycosidases commonly used for N- and O-linked glycan degradation [69] [70] [71] [72] [73] [74] [75] [76] [77] [78] [79] [80] [81] [82]. These evidence together with the previous partial structural and se-quence analyses of g200.
Also, we have extended references that provided the evidence the presence of GN1 epitope in g580.
4) O- and N-carbohydrates should be in italics.
We have set O- and N-linked glycans in italic form on page 4 lines 166 and 189, and page 18 lines 672 and 678.